# Frequency of Diarrheagenic Virulence Genes and Characteristics in *Escherichia coli* Isolates from Pigs with Diarrhea in China

**DOI:** 10.3390/microorganisms7090308

**Published:** 2019-09-02

**Authors:** Gui-Yan Yang, Liang Guo, Jin-Hui Su, Yao-Hong Zhu, Lian-Guo Jiao, Jiu-Feng Wang

**Affiliations:** College of Veterinary Medicine, China Agricultural University, Beijing 100193, China

**Keywords:** *Escherichia coli*, diarrhea, virulence gene, genetic diversity, antibiotic resistance, pig

## Abstract

Intestinal pathogenic *Escherichia coli* (InPEC) is a leading cause of postweaning diarrhea (PWD) in pigs. Here, a total of 455 *E. coli* strains were isolated from small intestinal content or feces from pigs with PWD in 56 large-scale (>500 sows; 10,000 animals per year) swine farms between 2014 and 2016. The frequency of occurrence of selected virulence factors for InPEC pathotypes was detected in 455 isolates by real-time PCR. Sequence types (STs), pulsed-field gel electrophoresis (PFGE), and antimicrobial susceptibility profiles of 171 *E. coli* isolates from 56 swine farms were further determined. The heat-labile enterotoxin (LT) was the most common (61.76%), followed by heat-stable enterotoxin (STb) (33.19%), *stx2e* (21.54%), STa (15.00%), *eae* (8.98%), *cnf2* (5.71%), *stx2* (5.71%), F18 (3.25%), and F4 (2.25%) with rates varying by geographic area and year of isolation. Notably, hybrids of *E. coli* isolates were potentially more virulent, as some InPEC hybrids (virotype F18:LT:*eae*:*stx2e*) can rapidly cause cell death in vitro. Genotypic analysis revealed that the most prominent genotype was ST10 (12.87%). The PFGE patterns were heterogeneous but were not ST or virotype related. A total of 94.15% of isolates were multidrug-resistant, with average resistance rates ranging from 90.05% for nalidixic acid to 2.34% for meropenem. Our investigation contributes to establishing the etiology of diarrhea and developing intervention strategies against *E. coli*-associated diarrheal disease in the future.

## 1. Introduction

Pathogenic *Escherichia coli* is one of the most important causes of postweaning diarrhea (PWD) in pigs, and the high associated mortality is responsible for significant economic losses [1]. Intestinal pathogenic *E. coli* (InPEC) strains in pigs include enterotoxigenic *E. coli* (ETEC), enteropathogenic *E. coli* (EPEC), and Shiga toxin-producing *E. coli* (STEC; also known as verocytotoxigenic *E. coli* (VETC)) [1]. Some pathotypes of *E. coli* (e.g., EPEC and STEC) are also a major public health concern as they have low natural infectious doses and are transmitted through ubiquitous mediums, including food and water [2]. Human and animal pathogenic *E. coli* strains share a common genetic background, and diseases caused by each pathotype involve specific colonization and virulence factors [3].

The ETEC from pigs produce one or both of two enterotoxins, heat-labile enterotoxin (LT) and heat-stable enterotoxin (STa and STb). The gene encoding STb was at one time the most frequently detected enterotoxin gene in ETEC recovered from pigs with PWD in Denmark during 1999–2000 [4]. By contrast, STa from ETEC is usually associated with disease in neonatal animals [1]. In comparison with ETEC in humans, the most common adhesins on ETEC from PWD in pigs are fimbriae F4 and F18 [5]. The EPEC strains are defined as the locus of enterocyte effacement (LEE) pathogenicity island and *eae*–harboring diarrheagenic *E. coli* that can form attaching and effacing (A/E) lesions on intestinal cells but do not possess the *stx* gene [2]. Shiga toxin-producing *E. coli* can cause disease by producing Shiga-like toxin (Stx). In pigs, STEC with the genetic variant *stx2e* possess the ability to cause edema disease (ED) [6].

Interestingly, a certain percentage of porcine ETEC isolates from pigs with PWD or ED are *stx2e* positive [1]. The ETEC strains that produce Stx (verotoxin (VT)) are appropriately designated hybrid ETEC/STEC or ETEC/VTEC [1]. Virulence genes encoding Stx2 and ST Ia have also been detected in both human and animal STEC isolates [7]. Our group found that infection with the hybrid strain F4^+^ ETEC/VTEC/EPEC causes intestinal inflammation even in F4 receptor-negative pigs [8]. Although data are available regarding the occurrence of pathogenic *E. coli* in various animal hosts [9,10], only limited information is available about the occurrence and characteristics of different InPEC, especially *E. coli* hybrids, from pigs.

Antibiotic treatments (prophylactic, metaphylactic, and therapeutic) of piglets against PWD usually results in the establishment of resistant bacteria in the gut microbiota [11]. It seems that the frequency of antimicrobial resistance among *E. coli* isolates from pigs is higher than that of isolates from cattle and sheep in Great Britain [12]. Therefore, it is important to figure out the antibiotic resistance of pathogenic *E. coli* from pigs in Chinese swine farms. In addition, several maternal vaccines on the market are used to prevent PWD by vaccination of the sow. Inactivated *E. coli* with purified fimbriae including F4 (K88), F5 (K99) and F6 (987P) without enterotoxins were applied in the pregnant sow in China.

The aim of this study was to investigate the proportion of selected virulence factors and antimicrobial resistance in *E. coli* isolates from pigs with PWD in Chinese large-scale (>500 sows; 10,000 animals per year) swine farms, and examine the genetic relatedness of these specific pathotypes to better understand the geographic dissemination of diarrheagenic *E. coli* in pigs.

## 2. Materials and Methods

### 2.1. Bacterial Isolates

A total of 455 *E. coli* isolates from the small intestinal contents or feces of piglets 20–50 days of age with postweaning diarrhea were collected from 56 large-scale swine farms in 15 provinces in China over a 3 year period (2014–2016) (Table 1). The 56 swine farms were selected by convenience sampling (Appendix A).

Black colonies with metallic luster on eosin-methylene blue plates were obtained and grown in Luria–Bertani broth for 12 h at 37 °C with shaking. For each sample, a maximum of 5 colonies were chosen based on initial morphology. The isolates were confirmed as *E. coli* based on standard biochemical tests using an API-20E Microbial Identification Kit (bioMérieux, MarcyI’Etoile, France) and polymerase chain reaction (PCR) amplification of the gene encoding the universal stress protein *uspA* [13,14]. A maximum of three *E. coli* strains per sample were included in the sample collection.

### 2.2. Detection of Virulence Genes Using Real-time PCR

Genomic DNA of 455 *E. coli* isolates was extracted using a TIANamp Bacteria DNA kit (Tiangen Biotech Inc., Beijing, China) according to the manufacturer’s instructions. The real-time PCR assays were performed on an ABI 7500 system (Applied Biosystems, Foster City, CA, USA), as previously described [10]. The sequences of the primers used are listed in Table 2. The following *E. coli* strains were used as positive controls: *E. coli* CVCC 225 (ETEC; F4^+^ LT^+^ STb^+^) and CVCC1450 (EPEC; *eae*^+^) (both obtained from the China Veterinary Culture Collection Center), and *E. coli* CICC 10670 (STEC; *eae*^+^
*stx2*^+^) (obtained from the China Center of Industrial Culture Collection). Negative control reactions without template were included in each run.

To further determine the background of *E. coli* hybrids, we detected the virulence genes (*Tir*, *espA*, *escV*) associated with LEE pathogenicity island.

### 2.3. Strain Selection Strategy for Genotypic and Antimicrobial Analysis

A total of 171 *E. coli* isolates selected from different swine farms (one isolate per sample or animal, first 3 numbered isolates per farm; in the case of less than 3 isolates in the farm, all of the isolates were included) were used for further genotypic and antimicrobial analysis.

### 2.4. Multilocus Sequence Typing

Multilocus sequence typing (MLST) was performed for 171 *E. coli* using internal fragments of the following seven housekeeping genes: *adk*, *fumC*, *gyrB*, *icd*, *mdh*, *purA*, and *recA*, according to protocols available from the *E. coli* MLST database (http://mlst.warwick.ac.uk/mlst/dbs/Ecoli). A minimum spanning tree (MS) was constructed using BioNumerics software (version 6.6; Applied Maths, Kortrijk, Belgium).

### 2.5. Pulsed-Field Gel Electrophoresis

Pulsed-field gel electrophoresis (PFGE) was performed according to the standardized Centers for Disease Control and Prevention PulseNet protocol (http://www.pulsenetinternational.org). Genomic DNA of 171 *E. coli* isolates was digested with *Xba*I (Takara Bio Inc., Shiga, Japan). The PFGE banding patterns were analyzed according to the Dice similarity coefficient method using BioNumerics software (version 6.6; Applied Maths). Clustering was determined using an unweighted pair group method with an arithmetic mean with 1.5% band tolerance and optimization of 1.0%. Isolates were considered to belong to the same PFGE subtype based on a similarity coefficient (*F* value) > 85%.

### 2.6. Antimicrobial Susceptibility Testing

A total of 171 *E. coli* isolates from different pig farm locations were tested for susceptibility to 18 antimicrobial agents that constitute the major antimicrobial classes (*n* = 10) used in veterinary and human medicine, in line with the microdilution method recommended by Clinical and Laboratory Standards Institute (CLSI) M100-S26 and CLSI VET01-A4 [15,16,17]. *Escherichia coli* ATCC 25922 was used for quality control. Based on the minimal inhibitory concentration determined for each drug, the isolates were classified as “susceptible”, “intermediate”, or “resistant”. Multidrug-resistant strains were those resistant to three or more antimicrobial classes, and intermediate isolates were not included.

### 2.7. Hemolytic Activity Determination

The hybrid *E. coli* isolates from pigs with diarrhea were further cultured on blood agar containing 5% sheep blood and incubated at 37 °C for 24 h to determine hemolysis.

### 2.8. Adhesion Assay

For a more detailed analysis of the virulence in hybrids of *E. coli* isolates from piglets with diarrhea, an adhesion assay and cell death assay in porcine intestinal epithelial J2 cells (IPEC-J2, ACC701, DSMZ) were performed.

InPEC isolates (LT^+^
*eae*^+^
*stx2e*^+^ hybrids; *n* = 6) which possessed fimbriae F18 as well as α-hemolytic activity were tested for adherence on IPEC-J2 cells. Enterotoxigenic *E. coli* (CVCC 225), EPEC (CVCC1450), and STEC (CICC 10670) were used as the positive control. The IPEC-J2 cells cultured with PBS (CONT) were used as a negative control.

Briefly, IPEC-J2 (1 × 10^5^ cells per well) were seeded onto 6 well plates (Gibco, Grand Island, NY, USA). On day 2, confluent monolayers of IPEC-J2 cells were infected with bacteria (2 × 10^6^ CFU/mL) at a multiplicity of infection (MOI) of 20. At 3 h after *E. coli* challenge, the IPEC-J2 monolayer was washed 3 times with PBS to remove non-adherent bacteria and then harvested by treatment with 0.05% trypsin for 10 min at 37 °C and plated on LB agar after serial dilution. Plates were incubated overnight at 37 °C, after which the number of CFUs was determined. The adherent CFU/mL of *E. coli* are presented.

The infected cell culture was also stained with 1% Giemsa (Shanghai Solarbio Bioscience &Technology Co., Ltd., Shanghai, China) as previously described [18]. Images were captured using an Olympus BX41 microscope (Olympus, Tokyo, Japan) equipped with a Canon EOS 550D camera head (Canon, Tokyo, Japan).

### 2.9. Cell Death Assay

The damage to the plasma membrane of IPEC-J2 under different conditions was quantified by the amount of lactate dehydrogenase (LDH) released using a LDH assay kit (Promega, Madison, WI, USA) according to the manufacturer’s instructions. The LDH activity in the supernatant 3 h following *E. coli* challenge was measured by monitoring the absorbance at 490 nm. The result was calculated using the following equation: ((LDH_infected_ – LDH_uninfected_)/(LDH_totallysis_ − LDH_uninfected_)) × 100.

### 2.10. Statistical Analysis

All statistical analyses were conducted using SPSS statistical software (version 19.0; SPSS Inc., Chicago, IL, USA). The Chi-square test was used to compare the frequency of occurrence of InPEC virulence genes in various provinces and year of isolation. The differences among groups in the adhesion and cell death assay were assessed using Tukey’s test. Data are presented as the mean ± standard error of the mean (SEM). A *p*-value < 0.05 was considered indicative of statistical significance.

## 3. Results

### 3.1. Prevalence of Virulence Genes in E. coli Isolates from Pigs with Diarrhea

The geographical distribution of *E. coli* used is shown in Appendix A. In this study, the most prevalent virulence factor of InPEC in piglets between 2014 and 2016 in Chinese large-scale swine farms was the ETEC-characteristic LT (61.76%) (Figure 1A). Heat-stable enterotoxin b (33.19%) and STa (15.00%), another two virulence factors specific for ETEC, occurred much less frequently than LT. A total of 113 of the 455 isolates (24.84%) were positive for both LT and STb. Only 3.25% and 2.25% of the 455 isolates were positive for F18 and F4, respectively. The intimin gene *eae* was detected in 8.98% of the 455 *E. coli* isolates examined. By contrast, *stx2*-positive isolates (5.71%) were much less prevalent. In particular, 21.54% of the *E. coli* isolates were *stx2e* positive. A total of 3.30% of the isolates were positive for both *eae* and *stx2*. Cytotoxic necrotizing factor type2 (*cnf2*), which is associated with diarrhea and septicemia, was detected in 5.71% of *E. coli* isolates from pigs with diarrhea (Figure 1A). Interestingly, a number of *E. coli* hybrids (3.96%) with specific genetic combination were identified in our study (Figure 3A).

Notably, 93 *E. coli* isolates in 2014 were all positive for LT, whereas the STb-positive isolates were detected more frequently in 2015–2016 than in 2014 (Figure 1B). The distribution of *E. coli* isolates from different farm locations varied in our study (Figure 1C).

### 3.2. Phylogenetic Analysis of Diarrheagenic E. coli Using MLST

A total of 171 *E. coli* isolates and three control strains (i.e., ETEC, EPEC, STEC) were analyzed using MLST, resulting in the identification of 65 sequence types (STs). Sequence type 10 (22 isolates) was the most common, followed by ST48 (16 isolates), ST29 (8 isolates), ST744 (8 isolates), ST101 (7 isolates), ST4214 (7 isolates), and ST617 (6 isolates). Four STs (i.e., ST100, ST165, ST410, and ST453) were detected in 5 isolates each, whereas ST88, ST361, and ST542 were detected in three isolates each, and ST46, ST77, ST117, ST189, ST209, ST1684, and ST3944 were detected in two isolates each. A total of 42 STs were detected only once (Appendix A).

Minimum-spanning trees showed that the tested *E. coli* mainly classed into five clonal complexes, which were represented by ST10, ST48, ST746, ST1437, and ST485, respectively. Sequence type 10 served as the predicted founder in the MS tree (Figure 2). Different STs were observed in each province except for Gansu province, whereas the provinces with the same ST were not all adjacent (Figure 2A, Appendix A). The InPEC strains that evolved from ST48 and ST746 were isolated in 2015 and 2016 but not 2014 (Figure 2B). Multiple virulence factors were expressed by isolates with the same kind of STs (Appendix A).

### 3.3. Phylogenetic Analysis of Diarrheagenic E. coli Using PFGE

In addition to genotype by MLST, the same *E. coli* isolates were further genotyped using PFGE. The PFGE typing of 171 *E. coli* isolates using *Xba*I led to their categorization into 38 clusters and 117 PFGE subtypes (data not shown). Most isolates including ETEC/EPEC, ETEC/STEC, and ETEC/ Necrotoxigenic *E. coli* (NTEC) hybrids showed unique PFGE patterns, and high heterogeneity was demonstrated within isolates of the same ST or virotype (Figure 3).

### 3.4. Antimicrobial Susceptibility

Multidrug resistance was observed in 94.15% of the 171 *E. coli* isolates. Resistance to nalidixic acid was the most common (90.05%), followed by resistance to trimethoprim-sulfamethoxazole (86.55%), ampicillin (84.80%), amoxicillin-clavulanate (84.63%), tetracycline (83.63%), florfenicol (77.78%), chloramphenicol (76.61%), enrofloxacin (72.51%), kanamycin (63.74%), cefazolin (60.82%), ciprofloxacin (60.82%), gentamicin (57.31%), ceftiofur (52.63%), streptomycin (40.35%), olaquindox (39.77%), polymyxin B (20.47%), amikacin (15.20%), and nitrofurantoin (2.34%) (Figure 4A). The proportion of multidrug-resistant isolates did not decline between 2014 and 2016 (Figure 4B).

### 3.5. Adherence and Cytotoxicity of Intestinal Pathogenic E. coli Hybrids to Intestinal Epithelial cells

The adherence phenotype of hybrid InPEC to IPEC-J2 cells was studied at 3 h after *E. coli* challenge, and diffusely-adherent bacteria were observed in ETEC and *E. coli* 15103 (Figure 5A). The adhesion rate of InPEC to IPEC-J2 cells was strain-dependent (Figure 5B). The number of adherent bacteria for *E. coli* 15103 was higher than STEC at 3 h after challenge (*p* < 0.01), whereas lower number of adherent bacteria was observed for the other InPEC isolates than both ETEC and EPEC (*p* < 0.05). Notably, most of the *E. coli* hybrids caused enormous cell damage to IPEC-J2 cells in this study (Figure 5C), we then measured the cell death induced by *E. coli* hybrids in vitro. As expected, the percentage of dead cells was higher after *E. coli* 15103 or 15104 infection than EPEC and STEC (*p* < 0.01; Figure 5D). The IPEC-J2 cell death induced by *E. coli* 16060 was more severe than STEC (*p* < 0.05).

## 4. Discussion

Enterotoxigenic *E. coli* is a primary cause of PWD (watery diarrhea) in pigs [1] and a major cause of traveler’s diarrhea in humans, particularly in developing countries [19]. As expected, our investigation revealed that heat-labile toxin (LT) was the most prevalent virulence factor among intestinal pathogenic *E. coli* from pigs with PWD in Chinese large-scale pig farms examined between 2014 and 2016. Remarkably, a number of hybrids of *E. coli* strains (e.g., LT^+^
*eae*^+^
*stx2*^+^) are present in newly weaned pigs with diarrhea.

Similarly, the relative proportions of LT, STb, and LT/STb toxin-producing ETEC seem to vary from one geographic area to another in patients with ETEC diarrhea [19]. Heat-labile toxin is a more significant contributor than STb to the virulence of F4^+^ ETEC infections in young F4ac receptor-positive pigs less than 2 weeks old [20]. Heat-stable toxin-ETEC diarrhea is more frequent in the summer, whereas LT-ETEC are present year-round and do not show seasonality [21]. According to our study, LT-positive *E. coli* (61.76%) appeared to be more common than *E. coli-*produced STb (33.19%) or LT/STb (24.84%). By contrast, STb (41/206 isolates, 20%) was more frequent than LT (23/206 isolates, 11%) in *E. coli* isolates from suckling pigs with diarrhea in China [22]. Among 215 *E. coli* isolates from pigs with PWD from eight provinces in eastern China, most isolates were positive for the heat-stable enterotoxin but not LT [23]. There was a positive relationship between fimbrial adhesin and enterotoxin genes which showed F4-positive *E. coli* usually produced both LT and STb [24]. However, most of the isolates that carry enterotoxin genes did not carry fimbrial adhesin genes (F4 and/or F18) in our study. The plasmids encoding F4 and/or F18 may be lost during storage of the strains or these fimbria-negative isolates possessed other fimbrial/non-fimbrial adhesin genes [25]. It remains to be determined in the future. Porcine EPEC, the second type of *E. coli* that is implicated in PWD, was found in about 8.98% of cases of PWD. In comparison, only 3.90% (8/206) of *eae*-harboring pathogenic *E. coli* isolates were detected from suckling pigs with diarrhea in China [22]. It has been concluded that porcine EPEC are included in members of the A/E pathogen family [2]. The formation of A/E lesions requires intimin which is encoded by the *eae* gene, and the other elements are encoded on the LEE [2]. However, intimin–Tir interaction was recently showed to be not sufficient for bacterial attachment to intestinal epithelial cells [26].

Of 455 porcine *E. coli* isolates examined in our study, 3.30% were positive for both *eae* and *stx2*. The detection rate of *stx2* and the subtype *stx2e* gene was different among *E. coli* isolates from pigs with diarrhea. Similar to other studies, 64% (436/687) and 38% (261/687) of 687 *E. coli* isolates from swine fecal samples were positive for the *stx2* and *stx2e* genes, respectively, in the United States [27]. In contrast to a previous report where only 6% (13/215) of the postweaning *E. coli* strains possessed the genes of *stx2e* [23], we found 21.54% (98/455) of the *E. coli* isolates were *stx2e*-positive from pigs with PWD. Unexpectedly [23], some *stx2e*-positive *E. coli* isolates were also positive for fimbrial F18 in our survey. It was reported that F18ab^+^ STEC were the prevalent pathogens of ED, and F18^+^ and/or intimin^+^ STEC/ETEC were the dominant pathogens of ED/PWD [28]. Shiga-like toxin 2 binds to the intestinal epithelium, where it induces further tissue damage [29]. The Stx2e-producing *E. coli* isolates from pigs and humans differ in virulence determinants and interactions with intestinal epithelial cells [30].

In addition to these three categories of diarrheagenic *E. coli* mentioned above (i.e., ETEC, EPEC, and STEC), necrotoxic *E. coli*, originally defined as strains of *E. coli* producing the toxin cnf, also cause diarrhea [14]. We found that 5.71% of porcine *E. coli* isolates were positive for the usually plasmid-encoded *cnf2* gene, which is associated with diarrhea and necrosis [31]. The *E. coli* strains that cause diarrhea have evolved by the acquisition, through horizontal gene transfer, of a particular set of characteristics that have successfully persisted in the host [32]. Shiga toxin-producing *E. coli* and other diarrheagenic *E. coli* can acquire virulence genes via horizontal gene transfer from other pathogroups leading to the development of intermediate or hybrid pathogroups [33]. The Stx-phages are able to transfer genes horizontally [33]. Enterotoxigenic *E. coli* harbors mobile genetic elements (such as plasmids and transposons), and plasmids coding for enterotoxin (e.g., LT, STb) and antibiotic resistance are frequently transferred together [32]. The usage of antibiotics may potentially promote the transfer of virulence genes between bacteria [34], but further research is needed in this field.

To better understand the genetic relatedness of these *E. coli* isolates, MLST and PFGE were performed. It has been shown that human and animal pathogenic *E. coli* strains share common genetic backgrounds [3]. In our study, the most frequently detected ST from diarrheal pigs was ST10 (22/171, 12.87%), which was indicated as being associated with the ancestral lineage of porcine *E. coli*. Sequence type 48 was the second most frequently detected ST (16/171, 9.36%). Consistent with these observations, ST10 and ST48 were found to be among the major clones in quinolone-resistant *E. coli* isolates from humans, animals, and the environment in the Czech Republic [35]. Recently, *E. coli* ST10 was isolated from human infections in China [36]. Sequence type 10, ST48, and ST29 were detected among atypical EPEC isolates from diarrhea patients and healthy carriers [37]. The genetic differences between the primary ST10 strains in mediating infection phenotypes remain to be defined.

The results of MLST and the PFGE-based clustering were reportedly consistent for *E. coli* O116 and OSB9 strains isolated from diseased swine in Japan [38]. Pulsed-field gel electrophoresis demonstrated that STEC isolates from pigs raised in the same finishing barn tend to be closely related [6]. We found that *E. coli* of the same ST and virotype generally exhibited identical PFGE profiles, whereas there was a high degree of genetic variability among specific InPEC with the same virotype or STs alone. Similarly, PFGE patterns were heterogeneous among *E. coli* strains from patients with Crohn’s disease, even strain from the same geographic origin [39]. Taken together, our data suggest that PFGE patterns of diarrheagenic *E. coli* are not highly correlated with virotype or STs.

Antibiotic treatment (ampicillin or tetracycline) does not appear to affect the genetic diversity of *E. coli* strains in pigs [40]. Shiga toxin-producing *E. coli* isolates of the same STs from healthy pigs in China generally exhibit the same or similar drug resistance patterns [41]. Nevertheless, in the present study, no strong correlation was observed between STs and resistance/virulence patterns for *E. coli* isolates from the same geographic area and the same year. Antibiotic ampicillin, amoxicillin–clavulanate, tetracycline, florfenicol, enrofloxacin, kanamycin, gentamicin, ceftiofur, and streptomycin were found to be used in Chinese large-scale swine farms [42]. The application of these antibiotics led to the emergence of multidrug-resistant InPEC. Therefore, non-antibiotic alternatives used to treat or prevent diarrheal disease in piglets is in urgent need. Probiotics, such as *E. coli* Nissle, are well-known and can be used to treat *E. coli* infections [43,44]. The genetic background of the host, which has a key role in driving the settlement of the gut microbiota, makes it possible to prevent *E. coli*-induced diarrhea in pigs by selective breeding for those are resistant to pathogenic *E. coli* [11]. We recently found that an uncommon hybrid strain of virotype F4ac:LT:STb:*stx2e*:*eae* (i.e., F4^+^ ETEC/VTEC/EPEC) causes intestinal inflammation in newly weaned pigs of the genotype *MUC4* that is supposed to be enterotoxigenic *E. coli* F4ab/ac receptor negative [8,45]. It has been pointed out that some *E. coli* isolates combine the main virulence characteristics of different pathotypes and are potentially more virulent hybrid pathogens [2]. From our result, LT^+^
*eae*^+^
*stx2e*^+^ F18^+^
*E. coli* hybrids could induce severe cell death in an IPEC-J2 cell infection model and 3.96% (18 of 455) of *E. coli* hybrids were isolated from pigs with PWD. Pathogenic *E. coli* need to attach to intestinal epithelial cells as the initial stage to cause cell damage [46]. The diffusely adherent phenomenon of *E. coli* was found in some F18^+^
*E. coli* isolates in this study. However, the adherence mechanisms of hybrid strains of *E. coli* pathotypes remain to be studied.

## 5. Conclusions

There is a high frequency of *E. coli* isolates carrying enterotoxin genes (LT, STb, and STa) but not F4 or F18 fimbrial genes from piglets with diarrhea. Genetically, InPEC show a high degree of genetic diversity in terms of PFGE patterns, even for strains of the same virotype or ST. Some hybrids are potentially more virulent such as LT^+^
*eae*^+^
*stx2e*^+^ F18^+^
*E. coli* isolates. Our study enhances understanding of the intestinal pathotypes of *E. coli* among diarrheal piglets in different geographic regions in China. Since these isolates typically are multidrug resistant to antibiotics, our data also provide potentially relevant specific virulence factors in achieving target product profiles for a future *E. coli* vaccine development to prevent diarrhea in pigs.

## Figures and Tables

**Figure 1 microorganisms-07-00308-f001:**
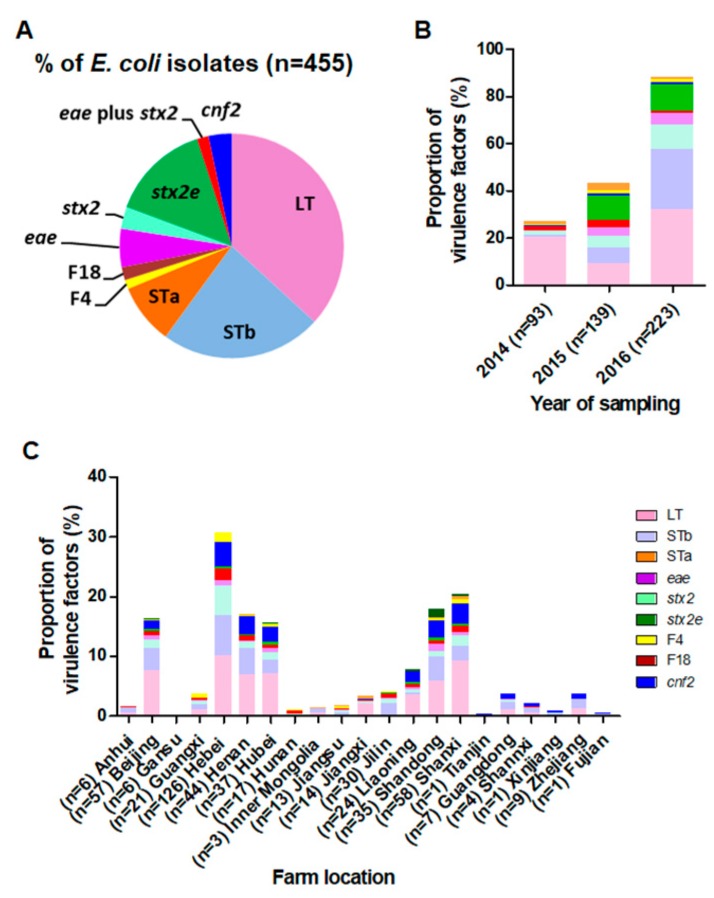
Frequency of virulence genes in *E. coli* isolates from pigs with diarrhea in China. (**A**) The occurrence of diarrheagenic virulence genes among the 455 *E. coli* isolates obtained from 2014 to 2016 in Chinese large-scale swine farms. Prevalence of virulence factors by (**B**) year of sampling and (**C**) sampling area.

**Figure 2 microorganisms-07-00308-f002:**
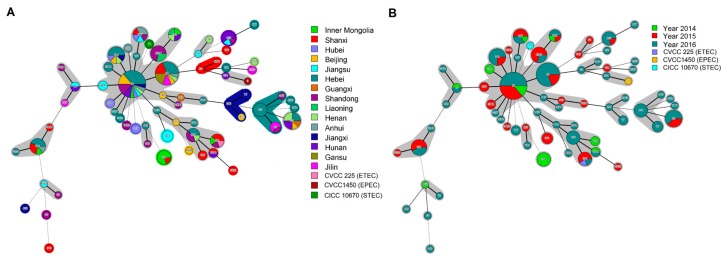
Minimum spanning trees showing the results of the multilocus sequence typing (MLST) of *E. coli* isolates in relation to geographic distribution and year of isolation. Three reference strains were included in the analysis. Each circle represents an ST and the size of the circle indicates the number of isolates of that particular type. The numbers in the circles represent the ST number. Solid lines denote the branch style coding up to 2, 3, and 4; dashed lines denote branch style coding up to 5; and dotted lines denote branch style coding above 5. The colors of the circles in the MLST trees indicate the MLST clonal complexes by province (**A**) and year of sampling (**B**).

**Figure 3 microorganisms-07-00308-f003:**
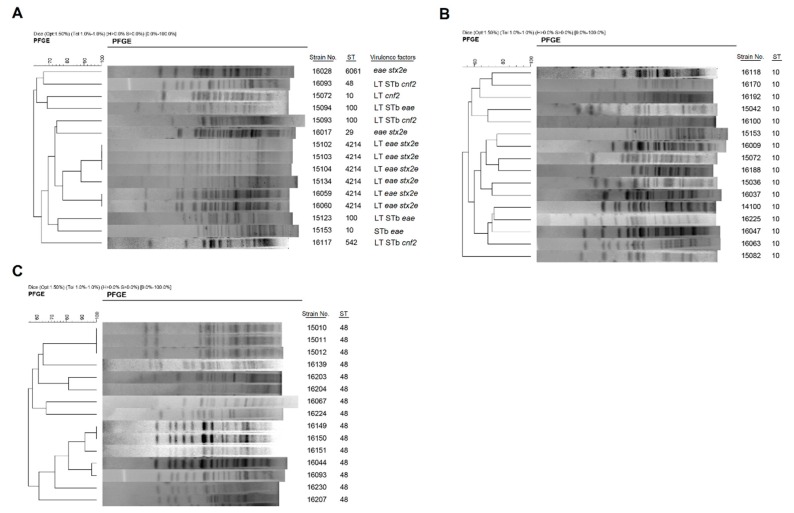
Pulsed-field gel electrophoresis (PFGE) of *Xba*I-digested DNA from *E. coli* hybrids. PFGE patterns, sequence types (STs) and virulence factors (VFs) of (**A**) hybrid *E. coli* strains, and (**B**,**C**) PFGE patterns of *E. coli* isolates with the same STs are shown.

**Figure 4 microorganisms-07-00308-f004:**
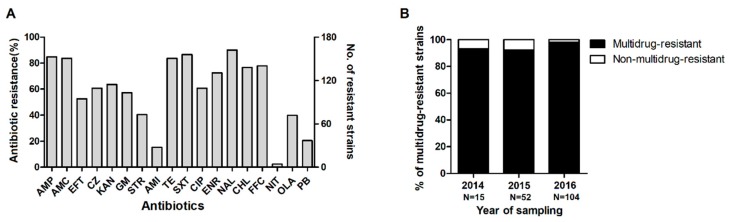
Resistance of the *E. coli* isolates from piglets with diarrhea to various antimicrobial agents. (**A**) The frequency of antibiotic resistance and the number of resistant strains among intestinal pathogenic *E. coli* from Chinese large-scale swine farms. (**B**) Proportion of multidrug-resistant strains between 2014 and 2016. The results of the tests for ampicillin (AMP), amoxicillin–clavulanate (AMC), ceftiofur (EFT), cefazolin (CZ), kanamycin (KAN), gentamicin (GM), streptomycin (STR), amikacin (AMI), tetracycline (TE), trimethoprim–sulfamethoxazole (SXT), ciprofloxacin (CIP), enrofloxacin (ENR), nalidixic acid (NAL), chloramphenicol (CHL), florfenicol (FFC), nitrofurantoin (NIT), olaquindox (OLA), and polymyxin B (PB) were interpreted according to CLSIM100-S26 and CLSI VET01-A4 guidelines. Multidrug-resistant isolates were those resistant to three or more antimicrobial classes, and intermediate isolates were not included.

**Figure 5 microorganisms-07-00308-f005:**
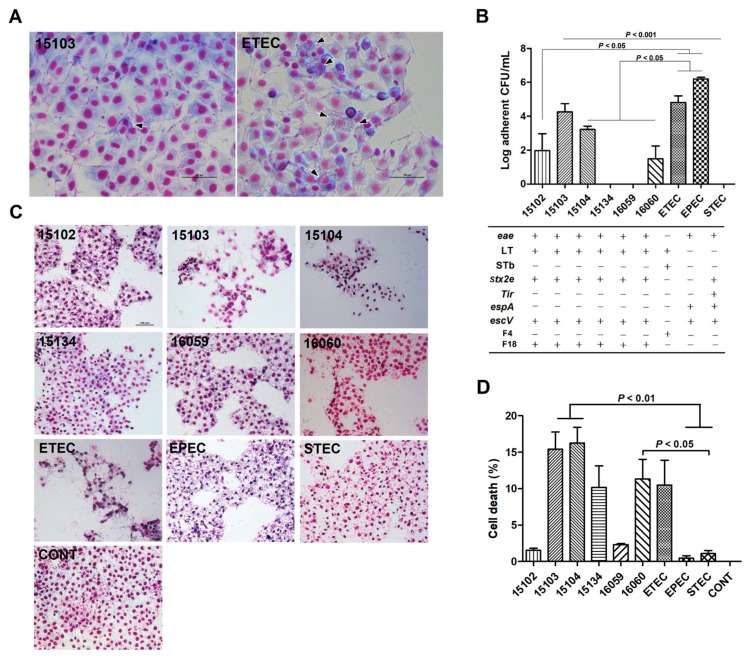
Adherence and cytotoxicity of *E. coli* hybrid strains to intestinal epithelial cells. The representative adherent phenotype (**A**) and adherence rate (**B**) of 6 *E. coli* hybrid isolates on porcine intestinal epithelial J2 cells (IPEC-J2) cells at 3 h after challenge are presented. Enterotoxigenic *E. coli* (CVCC 225), enteropathogenic *E. coli* (CVCC1450) and Shiga toxin-producing *E. coli* (CICC 10670) were used as positive controls, and PBS (CONT) was used as a negative control. The release of LDH was measured in supernatant collected from the indicated IPEC-J2 cell cultures at 3 h after *E. coli* challenge. The monolayers were washed, fixed, and stained with Giemsa. Scale bars, 100 µm. Data are presented as the mean ± SEM of 3 independent experiments.

**Table 1 microorganisms-07-00308-t001:** Month, year, and farm location of sample collections from piglets with diarrhea.

Sources (Number of Isolates)	Farm location (Number of Isolates)	Month	Year	Total Number of Isolates
Small intestine (93)	Shanxi (26), Hubei (24), Beijing (21), Liaoning (11), Hebei (11)	February to December	2014	93
Small intestine (139)	Hebei (37), Guangxi (21), Jilin (19), Shanxi (17), Liaoning (13), Jiangsu (13), Shandong (12), Henan (7)	January to December	2015	139
Small intestine (121), feces (102)	Hebei (64), Henan (37), Beijing (20), Shandong (20), Hunan (17), Jiangxi (14), Hubei (13), Shanxi (12), Jilin (11), Anhui (6), Gansu (6), Inner Mongolia (3)	March to December	2016	223

**Table 2 microorganisms-07-00308-t002:** Sequences of oligonucleotide primers used for PCR, length of the respective PCR product, and gene accession number.

Gene Product ^*a*^	Primer		Product Size (bp)	Accession Number
	Direction ^*b*^	Sequence (5′→3′)		
*uspA*	F	CCGATACGCTGCCAATCAGT	884	CP006636.1
	R	ACGCAGACCGTAGGCCAGAT		
LT	F	TTCCCACCGGATCACCAA	62	KF733767.1
	R	CAACCTTGTGGTGCATGATGA		
STa	F	CAACTGAATCACTTGACTCTT	158	CP025841.1
	R	TTAATAACATCCAGCACAGG		
STb	F	ATGTAAATACCTACAACGGGTGAT	360	M35729
	R	TATTTGGGCGCCAAAGCATGCTCC		
*eae*	F	CATTGATCAGGATTTTTCTGGTGATA	102	Z11541
	R	CTCATGCGGAAATAGCCGTTA		
*stx2*	F	CCACATCGGTGTCTGTTATTAACC	93	X07865
	R	GGTCAAAACGCGCCTGATAG		
*stx2e*	F	ATACGATGACGCCGGAAGAC	291	U72191.1
	R	TCAGAAACGCTGCTGCTGTA		
*cnf2*	F	GCGGAAATTTGAGCGGTTGT	165	U01097.1
	R	CGCGCGGCATTGGATTATTT		
*Tir*	F	GTTGGCTTTGACACCGGAAC	379	AF022236
	R	TACACCAGCACCAATTCCCC		
*espA*	F	TCAGAATCGCAGCCTGAAAA	60	AF022236
	R	CGAAGGATGAGGTGGTTAAGCT		
*escV*	F	ATTCTGGCTCTCTTCTTCTTTATGGCTG	544	AF022236
	R	CGTCCCCTTTTACAAACTTCATCGC		
F4	F	GAATCTGTCCGAGAATATCA	499	EU570252.1
	R	GTTGGTACAGGTCTTAATGG		
F18	F	TGGCACTGTAGGAGATACCATTCAGC	337	JX987521.1
	R	GGTTTGACCACCTTTCAGTTGAGCAG		

^*a*^*uspA*, universal stress protein; LT, heat-labile enterotoxin; ST, heat-stable enterotoxin; *eae*, *E. coli* attaching and effacing gene; *stx*, Shiga-like toxin; *cnf*, cytotoxic necrotizing factor; *Tir*, translocated intimin receptor; *espA*, *E. coli*-secreted protein A; F4, F4 fimbria. ^*b*^ F, forward; R, reverse.

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
