# Peer review of "Frequency of Diarrheagenic Virulence Genes and Characteristics in Escherichia coli Isolates from Pigs with Diarrhea in China"

_microorganisms, 2019, doi:10.3390/microorganisms7090308_

Round 1

Reviewer 1 Report

This study analyzed virulence, antimicrobial resistance profiles and molecular genetic typing of pathogenic E. coli isolates from piglets with postweaning diarrhea from 2014 to 2016 and confirmed cytotoxic effect of isolates through in vitro cell experiment. This manuscript is interesting, but some further studies on adhesins of the isolates would be informative to the readers. Also, author should do explain clearly the result and discussion of this study; especially in cytotoxicity and adherence of isolates. Before acceptance to publish, author should revise following lists of the major comments.

Pg 1, line 15: Antimicrobial susceptibility profiles of 171 E. coli

→ I think readers be curious why you investigate antimicrobial susceptibility for just 171
             of 455 E. coli isolates. Please write the reason for that.

Pg 1, line 19: hybrid E. coli

→ This term “hybrid E. coli” is not a frequently used word. Many scientists will be confused
because they do not know what this means.

Pg 1, line 38: The gene encoding STb was at one time the most frequently detected enterotoxin gene in ETEC recovered from pigs with PWD [4].

→ Frequencies of virulence gene are variable from area to area and over time in specific regions. Please write specific region and time where and when STb was the most frequently detected.

Pg 2, line 55: It seems that the frequency of antimicrobial resistance among commensal E. coli
isolates from pigs is higher than that 56 of isolates from cattle and sheep in Great Britain [12].

→ I think the aim of this study is to investigate the virulence factors and antimicrobial resistance profiles of E. coli from diarrheic piglets which means it is not from human, and not from commensal microbiota. This sentence clouds an argument, so should be revised.

Pg 2, line 71: Supplementary Fig. S1

→ Please check the data once again. In the paper, you said isolates were collected over a 3-year period (2014-2016), but supplementary data showed the data for 2014 to 2018.

Pg 2, line 75: API-20E Microbial Identification Kit (bioMérieux, Shanghai, China)

→ I think it should be revised as follows: (biomerieux, MarcyI'Etoile, France)

Pg 3, line 84: E. coli CVCC225 (ETEC), CVCC1450 (EPEC), CICC 10670 (STEC)

→ The virulence and antimicrobial resistance profiles of each positive controls are
valuable data for readers. Please write those data.

Pg 4, line 122: Hemolytic Activity Determination

→ There was no result about hemolytic activity of isolates in this study.

Pg 4, line 125: Adhesion Assay

→ Please write how many strains and what strains were analyzed for adhesion assay,
and why you selected those strains. In result section, there were only six test results.

Pg 5, line 173: Figure 1.

→ Unfortunately, the presentation of the results is rather difficult to follow. I also recommend replacing Figure 1 with charts.

Pg 6, line 193: Figure 2.

→ The resolution of this figure is too low for readers. Author should replace this figure with high-resolution.

Pg 6, line 203: ETEC/NTEC

→ Use of acronym without first being written out is not reader friendly. If you want to use acronym NTEC which means necrotoxigenic E. coli, you should first write the full name.

Pg 7, line 222: Figure 4 (B)

             → Just 15 isolates for 2014 is too small for representing the status of antimicrobial resistance in China. What is the authors’ explanation of this?

Pg 8, line 232: E. coli 15103 (Fig. 5A.)

→ I couldn’t find this strain in the section “Materials and Methods”. What is the E. coli 15103?

Pg 8, line 244: ETEC (CVCC 225), EPEC (CVCC1450) and STEC (CICC 10670) were used as controls.

→ Author should insert data of PBS as a negative control.

Pg 9, line 257: ST-ETEC diarrhea is more frequent in the summer, whereas LT-ETEC are present year round and do not show seasonality [20].

→ Compared to this reference, did your result show any seasonality in prevalence of pathotype, virulence factors or antimicrobial susceptibility?

Pg 9, line 264: However, most of the isolates that carry enterotoxin genes did not carry fimbrial adhesin genes in our study.

→ There are lots of adhesins, not only just F4 or F18. For the pathogenesis of E. coli, adhesin is essentially needed. Author should investigate another fimbrial / non-fimbrial adhesin genes, such as F5, F6, F41, paa, AIDA-1 etc.

Pg 9, line 269: lessions

→ lesions

Pg 9, line 275: 64 and 38% of 687 E. coli isolates from swine fecal samples

→ It is hard to understand for readers what the 64 and 38% is. This sentence should be revised to reader friendly.

Pg 9, line 279: Unexpectedly [22,26], some stx2e-positive E. coli isolates were also positive for fimbrial F18 in our survey 279

→ Frequency of virulence factors are variable depending on time and region. Reference 26 was published in 1994, so this data is hard to represent latest relationship between fimbria and pathotype. Also, this reference reported that there was relationship between F18 and STEC.

Pg 9, line 284: To determine how virulence factors affect the pathogenicity of porcine EPEC and Stx2e-producing E. coli requires further study.

→ This sentence may be controversial to many scientists. There were numerous studies reporting how and what virulence factors of EPEC and STEC affect the pathogenicity to pig’s intestine.

Pg 9, line 296: The usage of antibiotics may potentially promote the transfer of virulence genes between bacteria, but further research is needed in this field.

             → Reference should be inserted for this sentence.

Pg 10, line 320: Antibiotic ampicillin, amoxicillin-clavulanate, tetracycline, florfenicol, enrofloxacin, kanamycin, gentamicin, ceftiofur, and streptomycin were found to be used in Chinese large-scale swine farms.

→ Reference should be inserted for this sentence.

Pg 10, line 331: From our result, LT+ eae+ 331 stx2e+ F18+ E. coli hybrids could induce severe cell
death in an IPEC-J2 cell infection model.

→ It is interesting that isolate had severe pathogenicity to cell. The aim of this study is to “investigate the proportion of selected virulence factors and antimicrobial resistance in E. coli isolates from pigs with PWD in China, and examine the genetic relatedness to better understand the geographic dissemination of diarrheagenic E. coli in pigs.”. In this point, readers will have interest “how many that isolate be in China”. Author should emphasize this on discussion section.

Pg 10, line 337: There is a high frequency of the E. coli isolates that carry enterotoxin genes (LT, STb and STa) but not F4 or F18 fimbrial genes from piglets with diarrhea.

→ F4 and F18 is known to be most frequently detected fimbrial genes in weaned piglets, worldwide. The reason why F4 and/or F18 fimbrial genes were not detected in this study should be mentioned in discussion.

Author Response

Reviewer 1

Pg 1, line 15: Antimicrobial susceptibility profiles of 171 E. coli

→ I think readers be curious why you investigate antimicrobial susceptibility for just 171of 455 E. coli isolates. Please write the reason for that.

AU: We appreciate with the reviewer’s constructive suggestions. Considering the similarity among E. coli isolates in the same swine farm, a total of 171 E. coli isolates selected from different swine farms (one isolate per sample or animal, first 3 numberd isolates per farm; In the case of less than 3 isolates in the farm, then all of the isolates were included.) were used for further genotypic and antimicrobial analysis. It has been presented in the M&M section 2.3.

Pg 1, line 19: hybrid E. coli

→ This term “hybrid E. coli” is not a frequently used word. Many scientists will be confused. because they do not know what this means.

AU: Thank you for the comment. It has been revised as “hybrids of E. coli” isolates……It can be referred in Nyholm et al. 2014 and Nyholm et al. 2015.

Pg 1, line 38: The gene encoding STb was at one time the most frequently detected enterotoxin gene in ETEC recovered from pigs with PWD [4].

→ Frequencies of virulence gene are variable from area to area and over time in specific regions. Please write specific region and time where and when STb was the most frequently detected.

AU: Thank you for the important comment. We have carefully checked the references and added the information.

“The gene encoding STb was at one time the most frequently detected enterotoxin gene in ETEC recovered from pigs with PWD in Denmark during 1999-2000 [4]”

Pg 2, line 55: It seems that the frequency of antimicrobial resistance among commensal E. coli isolates from pigs is higher than that 56 of isolates from cattle and sheep in Great Britain [12].

→ I think the aim of this study is to investigate the virulence factors and antimicrobial resistance profiles of E. coli from diarrheic piglets which means it is not from human, and not from commensal microbiota. This sentence clouds an argument, so should be revised.

AU: The sentence has been revised as the reviewer suggested.

Pg 2, line 71: Supplementary Fig. S1

→ Please check the data once again. In the paper, you said isolates were collected over a 3-year period (2014-2016), but supplementary data showed the data for 2014 to 2018.

AU: Thank you for the comment. The supplementary data have been corrected as 2014-2016 (Supplementary Fig. S1). We have carefully checked the whole text and corrected.

Pg 2, line 75: API-20E Microbial Identification Kit (bioMérieux, Shanghai, China)

→ I think it should be revised as follows: (biomerieux, MarcyI'Etoile, France)

AU: It has been revised as suggested.

Pg 3, line 84: E. coli CVCC225 (ETEC), CVCC1450 (EPEC), CICC 10670 (STEC)

→ The virulence and antimicrobial resistance profiles of each positive controls are valuable data for readers. Please write those data.

AU: Thank you for the constructive comment. We have added the virulence profiles of these three strains in the revision as suggested (Line 89-91). Besides, some positive controls are strains isolated in this study.

As E. coli CVCC225 (ETEC), CVCC1450 (EPEC), CICC 10670 (STEC) are not reference strains in the antimicrobial resistance study, and our aim was to investigate the antimicrobial susceptibility of E. coli isolates, so antimicrobial resistance profiles of each positive controls were not added.

Pg 4, line 122: Hemolytic Activity Determination

→ There was no result about hemolytic activity of isolates in this study.

AU: InPEC isolates (LT+ eae+ stx2e+ hybrids) which possess fimbriae F18 as well as a-hemolytic activity were tested for adherence on IPEC-J2 cells (line 137-138).

Pg 4, line 125: Adhesion Assay

→ Please write how many strains and what strains were analyzed for adhesion assay, and why you selected those strains. In result section, there were only six test results.

AU: The number of E. coli isolates (n = 6) included in the in vitro studies was added in the revision.

“InPEC isolates (LT+ eae+ stx2e+ hybrids; n = 6) which possess fimbriae F18 as well as a-hemolytic activity were tested for adherence on IPEC-J2 cells.”

Pg 5, line 173: Figure 1.

→ Unfortunately, the presentation of the results is rather difficult to follow. I also recommend replacing Figure 1 with charts.

AU: Thank you for this good comment. Figure 1 might be replaced with 3 charts. However, figure maybe much better than a chart in this case. We have revised Figure 1 to make it more understandable.

Pg 6, line 193: Figure 2.

→ The resolution of this figure is too low for readers. Author should replace this figure with high-resolution.

AU: Figure 2 has been replaced with high-resolution.

Pg 6, line 203: ETEC/NTEC

→ Use of acronym without first being written out is not reader friendly. If you want to use acronym NTEC which means necrotoxigenic E. coli, you should first write the full name.

AU: It has been revised (line 213).

Pg 7, line 222: Figure 4 (B)

→ Just 15 isolates for 2014 is too small for representing the status of antimicrobial resistance in China. What is the authors’ explanation of this?

AU: Based on the strain selection strategy for genotypic and antimicrobial Analysis described in M&M 2.3 (approximately 3 isolates per farm), and five swine farms were included in the sample collection (Table 1).

Pg 8, line 232: E. coli 15103 (Fig. 5A.)

→ I couldn’t find this strain in the section “Materials and Methods”. What is the E. coli 15103?

AU: E. coli 15103 is an InPEC isolate (LT+ eae+ stx2e+) which possesses fimbriae F18 as well as a-hemolytic activity from pigs with PWD.

Pg 8, line 244: ETEC (CVCC 225), EPEC (CVCC1450) and STEC (CICC 10670) were used as controls.

→ Author should insert data of PBS as a negative control.

AU: Thank you for this good suggestion. We did use PBS as a negative control. The data have been added in Figure 5. The relative information has been added in the M&M section (line 139-140) and Figure 5 legend (line 253).

Pg 9, line 257: ST-ETEC diarrhea is more frequent in the summer, whereas LT-ETEC are present year round and do not show seasonality [20].

→ Compared to this reference, did your result show any seasonality in prevalence of pathotype, virulence factors or antimicrobial susceptibility?

AU: We appreciate for the reviewer’s concern. Our results did not show the seasonality in prevalence of pathotype, virulence factors or antimicrobial susceptibility. The ability of E. coli to survive in the open environment involves multiple factors, such as temperature, moisture and pH. These data thus indicate that LT-ETEC maybe more adapted to diverse environmental challenges than STb-ETEC in swine farms. It helps to explain why there was a higher prevalence of LT than STb.

Pg 9, line 264: However, most of the isolates that carry enterotoxin genes did not carry fimbrial adhesin genes in our study.

→ There are lots of adhesins, not only just F4 or F18. For the pathogenesis of E. coli, adhesin is essentially needed. Author should investigate another fimbrial / non-fimbrial adhesin genes, such as F5, F6, F41, paa, AIDA-1 etc.

AU: Thank you for providing us the valuable and constructive comments. The aim of the present study was to investigate the occurrence of relatively prevalent adhesin genes in E. coli pathotypes, especially ETEC. Multiple previous studies showed that fimbriae F4 and F18 were the most common adhesins on ETEC from PWD in pigs (Qadri et al., 2005 etc). According to our results, there was a low number of F4/F18-positive E. coli strains isolated from pigs with PWD. One speculation is that these fimbria-negative isolates possessed another fimbrial/non-fimbrial adhesin genes [24]. It remains to be determined in the future. We have rewritten this sentence and added the discussion in the revision.

Pg 9, line 269: lessions

→ lesions

AU: It has been corrected.

Pg 9, line 275: 64 and 38% of 687 E. coli isolates from swine fecal samples

→ It is hard to understand for readers what the 64 and 38% is. This sentence should be revised to reader friendly.

AU: This sentence has been revised. “64% (436 of 687) and 38% (261 of 687) of 687 E. coli isolates from swine fecal samples were positive for the stx2 and stx2e genes, respectively, in the United States [25]”

Pg 9, line 279: Unexpectedly [22,26], some stx2e-positive E. coli isolates were also positive for fimbrial F18 in our survey 279

→ Frequency of virulence factors are variable depending on time and region. Reference 26 was published in 1994, so this data is hard to represent latest relationship between fimbria and pathotype. Also, this reference reported that there was relationship between F18 and STEC.

AU: We agree with the reviewer’s comment. Reference 26 has been deleted in the revision.

Pg 9, line 284: To determine how virulence factors affect the pathogenicity of porcine EPEC and Stx2e-producing E. coli requires further study.

→ This sentence may be controversial to many scientists. There were numerous studies reporting how and what virulence factors of EPEC and STEC affect the pathogenicity to pig’s intestine.

AU: As suggested, this sentence has been deleted in the revision.

Pg 9, line 296: The usage of antibiotics may potentially promote the transfer of virulence genes between bacteria, but further research is needed in this field.

→ Reference should be inserted for this sentence.

AU: Thank you for the important comments. Reference has been inserted in the revision.

“Martinez, J.L.; Baquero, F. Interactions among strategies associated with bacterial infection: pathogenicity, epidemicity, and antibiotic resistance. Clin. Microbiol. Rev. 2002, 15, 647-679.”

Pg 10, line 320: Antibiotic ampicillin, amoxicillin-clavulanate, tetracycline, florfenicol, enrofloxacin, kanamycin, gentamicin, ceftiofur, and streptomycin were found to be used in Chinese large-scale swine farms.

→ Reference should be inserted for this sentence.

AU: Thank you for this good suggestion. According to our survey and the other studies, antibiotic ampicillin, amoxicillin-clavulanate, tetracycline, florfenicol, enrofloxacin, kanamycin, gentamicin, ceftiofur, and streptomycin were found to be used in Chinese large-scale swine farms. Reference has been inserted for this sentence (Zhu, Y.G.; Johnson, T.A.; Su, J.Q.; Qiao, M.; Guo, G.X.; Stedtfeld, R.D.; Hashsham, S.A.; Tiedje, J.M. Diverse and abundant antibiotic resistance genes in Chinese swine farms. Proc. Natl. Acad. Sci. USA. 2013, 110, 3435-3440).

Pg 10, line 331: From our result, LT+ eae+ 331 stx2e+ F18+ E. coli hybrids could induce severe cell death in an IPEC-J2 cell infection model.

→ It is interesting that isolate had severe pathogenicity to cell. The aim of this study is to “investigate the proportion of selected virulence factors and antimicrobial resistance in E. coli isolates from pigs with PWD in China, and examine the genetic relatedness to better understand the geographic dissemination of diarrheagenic E. coli in pigs.”. In this point, readers will have interest “how many that isolate be in China”. Author should emphasize this on discussion section.

AU: Thank you for the good recommendation. We have added the number of hybrids of E. coli isolates in the discussion section (line 346-347).

Pg 10, line 337: There is a high frequency of the E. coli isolates that carry enterotoxin genes (LT, STb and STa) but not F4 or F18 fimbrial genes from piglets with diarrhea.

→ F4 and F18 is known to be most frequently detected fimbrial genes in weaned piglets, worldwide. The reason why F4 and/or F18 fimbrial genes were not detected in this study should be mentioned in discussion.

AU: We appreciate for your important comments. In the discussion section, we have added some explanations about why there is a low number of F4/F18-positive E. coli strains isolated from pigs with PWD in the present study.

“The plasmids encoding F4 and/or F18 may be lost during storage of the strains or these fimbria-negative isolates possessed another fimbrial/non-fimbrial adhesin genes [24]. It remains to be determined in the future.”

Reviewer 2 Report

Review of manuscript entitled: Frequency of diarrheagenic virulence genes and characteristics in Escherichia coli isolates from pigs with diarrhea in China

The data obtained are valuable and the analysis performed by the authors is informative and educational for the audience of the journal Microorganisms. The article conforms to the journal-specific instructions. The title accurately reflects the content of the manuscript and prompt the audience to its reading. The abstract is concise. The methodology is described in enough detail and adequate, and the conclusions support the evidence of finding different to prevent diarrhoea in pigs. Statistics used in the study are also adequate. This study provides a convincing contribution to a better understanding of the etiology of diarrhoea in pigs. Last but not list, most importantly the data are presented and discussed in a very balanced way, which in science it is an important factor that enhances the quality of the manuscript and its conclusions. The English language and style are fine and only a minor spell check is necessary.

Suggestions:

Line 55 instead of using the term ‘microflora’ I suggest using ‘microbiota’ as this term is more correct in light of the sentence on page 324: ‘’Therefore, nonantibiotic alternatives used to treat or prevent diarrheal disease in piglets is in urgent need’’ I propose to add the possibility to suggest using probiotics, ie. coli Nissle 1917 to establish , as it is a well-known probiotic that can be applied for various treatments of the intestinal tract of mammals referenced from: (Study of the In Vitro Antagonistic Activity of Various Single-Strain and Multi-Strain Probiotics against Escherichia coli Int J Environ Res Public Health, 2018 Jul; 15(7): 1539 and Antagonistic effects of probiotic Escherichia coli Nissle 1917 on EHEC strains of serotype O104:H4 and O157:H7. Int. J. Med. Microbiol. 2013;303:1–8.

Author Response

Line 55 instead of using the term ‘microflora’ I suggest using ‘microbiota’ as this term is more correct in light of the sentence on page 324: ‘’Therefore, nonantibiotic alternatives used to treat or prevent diarrheal disease in piglets is in urgent need’’ I propose to add the possibility to suggest using probiotics, ie. coli Nissle 1917 to establish , as it is a well-known probiotic that can be applied for various treatments of the intestinal tract of mammals referenced from: (Study of the In Vitro Antagonistic Activity of Various Single-Strain and Multi-Strain Probiotics against Escherichia coli Int J Environ Res Public Health, 2018 Jul; 15(7): 1539 and Antagonistic effects of probiotic Escherichia coli Nissle 1917 on EHEC strains of serotype O104:H4 and O157:H7. Int. J. Med. Microbiol. 2013;303:1–8.

AU: As suggested,‘microflora’ has been replaced by ‘microbiota’.

Thank you for the good suggestion. The following sentence has been added in the revision. “Probiotic, such as E. coli Nissle, is well-known that can be used to treat E. coli infections [41,42]” line 337-338

Round 2

Reviewer 1 Report

This data will be valuable for understanding the latest virulence characteristics of Escherichia coli from pigs, and establishing treatment and prevention strategies for colibaillosis in the swine industry.